# Indonesian Traditional Market Flexibility Amidst State Promoted Market Competition

**Mangku Purnomo [1,*], Fenna Otten [2] and Heiko Faust [2]**

[1]  Department of Socio-Economics, Faculty of Agriculture, Brawijaya University, Jln. Veteran, Malang 65145, Indonesia

[2]  Department of Human Geography, University of Göttingen, 37077 Göttingen, Germany; fenna.otten@geo-uni-goettingen.de (F.O.); hfaust@gwdg.de (H.F.)

*   Correspondence: mangku@ub.ac.id; Tel.: +62-0341-580054

**Abstract:** The penetration of modern supermarkets is believed to be the cause of the declining role of traditional markets and street vendors in Indonesia. Nevertheless, the competition between state-promoted markets and traditional markets is rarely discussed, both adaptation of market institution and strategy of market actors. This research outlined a theoretical understanding of the dynamics of traditional markets, along the concepts of market flexibility as an adaptation strategy and coordination problems as market actor strategies. The researchers empirically reflect the strategies of four traditional vegetable markets that still survive from tight competition—both the market itself as a social institution, and the strategies of actors involved in market transactions. The traditional market builds flexibility by: (1) Specifying commodities, (2) segmenting customers, (3) changing market operating hour, (4) modifying transportation to operate more efficiently, and (5) low cost market management. At the actor level, competition problems are resolved by utilizing an emotional sentiment of friendship social relations; the formation of prices is determined by developing effective networks of information; and the cooperation problem is dealt with by building a system of punishment and reward based on informal mechanisms. This finding verifies the thesis stating that market competitiveness is determined by institutional flexibility against competition and the ability of market actors to build effective social interactions to maintain market sustainability. Based on the above explanation, further research needs to be focused on calculating how much efficiency is built due to market flexibility, both the transaction cost and the production cost in a quantitative manner. At the actor level, it is necessary to delineate the strategies being built, whether based on pure rational or economic and moral or non-economic considerations in solving coordination problems in the market.

**Keywords:** traditional market; state promoted market; market flexibility; market actors; competition; valuation; coordination

## 1. Introduction

Studies on the market these days focus on the problem of stability of market regularity/market order, as well as markets efficiency issues (Beckert 2010; Chowdhry and Nanda 1998; Dimitriadou et al. 2018; Fligstein 2002). Whilst markets as social institutions change over time, these changes have received little attention so far (Jackson 2003; Çalişkan and Callon 2009; Overdevest 2011). The discourse about traditional market transformation in developing countries, including Indonesia, focuses rather on the competition between traditional markets and supermarkets or street vendors, whereas the competition between markets, particularly between traditional and modern markets pioneered by the State, is rarely discussed (Roslin and Melewar 2008; Toiba et al. 2015;

Prabowo and Rahadi 2015). The modernization policy of traditional markets in Indonesia is currently focused on creating a new marketplace rather than the old market revitalization due to the large potential conflict between the government and traders. Because the conflict is considered to be a hindrance, the government prefers to build new markets to avoid social conflicts (Fahmi et al. 2016). It is also considered easier and more efficient on a budget basis.

Additionally, the literature on traditional market competition and the state promoted market is still very small, especially for the case of developing countries. The vast majority of research mainly focuses on competition between traditional markets and supermarkets (Huang et al. 2015), while competition with state promoted markets is rarely discussed, especially vegetable and fruit market. In fact, vegetable and fruit markets in Indonesia have experienced significant growth for last ten years, in line with an increase of vegetable consumption per capita and natural population growth (Shepherd 2004). According to the 2017 data from the Department of Agriculture, an average Indonesian consumes only 34.52 kg of fruits and 40.33 kg of vegetables annually. At the same time, the Food and Agriculture Organization (FAO) has recommended that each individual should consume at least 73 kg of fruits and 91.25 kg of vegetables every year (CBS 2018).With that goal in mind and a population approaching 250 million people, Indonesia is a huge market that will continue to grow (CBS 2018).

The increased consumption of vegetables can be seen from the emergence of new vegetable markets, mainly initiated by the government as a competitor to traditional vegetable markets. The new market appears at the provincial level called "Center of Agribusiness" (CA) and at the district level called "Sub-Center Agribusiness" (STA). In Indonesia, in particular Java Island, a province is a state-level government in charge of districts where each district has an average population of two-hundred thousand inhabitants with some exceptions. The district of Malang city, for example, has a population of more than 900 thousand inhabitants. Not only do TA and STA simplify the integration of the supply chain, but they also develop new markets as substitutes for the existing traditional wet markets as one of the strategies under the agropolitan program (Fatkhiati et al. 2015; Subadyo and Poerwoningsih 2017). By 2015, Sub-Terminal Agribusiness had been established within almost all districts in Indonesia, especially in areas of major vegetable production (Mariyono et al. 2017). The government subsidizes these markets directly and indirectly, which will threaten the existence of traditional markets (Anugrah 2004; Wardhana et al. 2017).

This article will specifically assess how the traditional vegetable market as a social institution flexibly transforms itself into a more efficient form to ensure they are able to compete with the state promoted market. Furthermore, at the actor level, the strategy to resolve coordination problems as a prerequisite for the transaction and market sustainability will also be described. Social interaction of actors to form a unique social network will also be described to determine the pattern of social relations in the market and to illustrate the process of encountering better market efficiency. Market flexibility concepts are applied to analyze how the concepts facilitate the social processes within markets to develop a one-of-a-kind adaptation method in order to solve potential issues. Once these issues are minimized, markets can carry out their function as social institutions and withstand amidst competition. The market actors develop strategies to ensure that transactions continue to sustain the market. The concept of market coordination problems will be the next main analytical framework. Taking a sample of four major traditional markets around the District of Malang, this article captures how they withstand and adapt to the new government-driven market dynamics in order to remain competitive.

## 2. Review of Literature

Markets are not just an economic mechanism for the allocation of goods, but are social institutions, inseparably interwoven with the political, social, and cultural environments in which they operate (Lie 1997; Smelser and Swedberg 2005; Beckert 2007; Fourcade 2007). Markets are a social structure for the exchange of rights, which enables people, firms and products to be evaluated and priced (Swedberg 2015). As the market is part of the overall social system, then it will continue to change as

an endeavor to adjust to the social changes. Therefore, the market will survive as a social institution if it is able to continue to change itself to adapt to consumer behavior, as well as competition with other markets.

An institutional approach that can be used to analyze how the market continues to adjust to external pressures and internal competition is the concept of market flexibility (Jackson 2015). Flexibility is the ability of a market as a social institution to transform and adapt to internal and external pressure related to changes in the behavior of actors, as well as values, so that market remains. Competition victory is not only the ability to improve transaction efficiency alone, but also its capability to change values, consumer segmentation, commodity changes, and social interaction changes, among others. "Flexibility" is understood as a way to analyze markets using a more realistic approach to eliminate external exposure through accommodating it, rather than refusing it (Jackson 2015). This approach will be very relevant to the transformation of traditional markets especially for developing countries where market stability is relatively weak and threatened with increasing external pressure (Purnomo 2018b).

Furthermore, as a social institution, markets can be seen as a living entity with an actor arena for mutual exchange, where the sustainability of the market highly depends on its ability to facilitate the exchange processes (Beckert 1996; Araujo 2007; Çalişkan and Callon 2010). Market actors develop social structures to mediate the problems they encounter with regard to exchange, competition, and production (Callon 1999). Through socially structuring the markets, it can be argued that social relationships underlying the markets have effects on their efficiency (Mawejje and Holden 2014). Theoretically, efficiency is a prerequisite of market sustainability and will only occur when market actors are capable of completing three problems of market coordination; namely the problem of competition, the problem of valuation, and the problem of cooperation (Beckert 2012).

Competition refers to the institutional and social structural devices in markets through which producers reduce uncertainties and secure profit opportunities. Market participants will look for the most efficient formula to keep acquiring maximum profit, often by taking the surplus of other actors, which incites even more competition (Beckert 2011, 2012; Beckert and Aspers 2011). Valuation refers to the creation of preferences in the markets and the classification of goods through firm and consumers. The basis for this is a process of classification and commensuration in which actors rank products according to their contribution to the fulfillment of a functional need within a certain status or order of goods (Aspers 2009; Beckert 2011; Beckert and Aspers 2011). The problem of cooperation refers to the social risks market actors face from the non-fulfillment of contracts due to the defection of their trading partners. Because of this problem, actors often build good relationships with other market players, who sometimes have nothing to do with the actual transaction process binding partners in order not to withdraw from the deal. Sometimes traders offer loans or specific facilities to ensure that trading partners do not cancel the deal. In this process, trust becomes an important component to assure the cooperation's compliance (Batt 2003; Brooks et al. 2017; Kwon and Suh 2004).

Based on the description above, the market (as a social institution) must be able to have strong flexibility to make changes to values, consumer segmentation, commodity changes, social interaction changes, efficiency as the "soul" of the market to be competitive. Market competitiveness is the key to the concept of markets flexibility (Jackson 2015). This research aimed to determine how the traditional vegetable market has changed to improve market efficiency amid competition with new markets in an institutional manner, especially those built by the government. How the market changes will be explained historically to ensure market institutional transformation can be explained in more detail. In order to determine the market actors' strategy to maintain transaction sustainability as the core of the social market process, the researchers approached the issue using the concept of "coordination problems".

Furthermore, to provide a broader perspective, the actors' strategies will be evaluated with the concept of five forces from the Portes (Lee et al. 2012; Wu et al. 2012). A threat of the new entrants is a barrier for new actors if they want to enter certain markets. Bargaining of suppliers is the strength that is owned by the seller to increase or decrease the price, which is very relative to the existence of other alternative markets. Meanwhile, the bargaining power of buyers is the buyer's strength to increase

prices or demand better quality. A threat of substitute whether certain products can be replaced by other products so that actors have more option. Finally, rivalry among existing competitors is also a barrier for new actors to enter the market.

## 3. Materials and Methods

This type of research fully uses a qualitative approach, so the first author and first co-author as researchers are the research instrument itself (Graneheim et al. 2017) where all field research is carried out by researchers independently. This research does not involve enumerators or other researchers. Therefore, the findings are the research result conducted by the author and co-author. The second co-author designed a research and data analysis theme and strategies, as well as formulated recommendations to ensure field research. Describing social processes in detail and historical dimensions, the qualitative approach can provide an in-depth and comprehensive picture of social phenomena that will be examined (Krefting 2017), and how markets and actors simultaneously develop strategies to deal with the competition.

The study was realized in the Malang region (with the regional capital of the same name), in the second largest city, in East Java. It is located south of Surabaya, the capital of East Java Province, Indonesia. Malang region is considered as an agricultural area characterized by rich water resources and fertile soils and flanked by three active volcanoes namely Arjuno, Kelud and Semeru. The region's land is ideal for the cultivation of agricultural products, such as rice, vegetables and fruit. Cabbage plants, for instance, contribute more than 25% of the total national production since 2010 (CBS 2018). Malang region supplies major cities in Indonesia, such as Jakarta, Surabaya and Yogyakarta, and other cities outside Java with vegetables and fruits. Some processed fruit and vegetable products are also exported to Taiwan, Japan, Korea and the Middle East. Large supermarkets located in eastern parts of Indonesia, such as Kalimantan, Sulawesi, and Papua also rely on Malang as a major supplier. More than 50 vegetable suppliers both for the inter-island market and supermarkets are domiciled in Malang (CBS 2018). They all belong to large wholesale markets because the buyers will resell to small traders and the retail market. The same applies to the four markets examined in this study; The Kedungboto market, the Karangploso market, the Gadang market and the Lawang market (Figure 1).

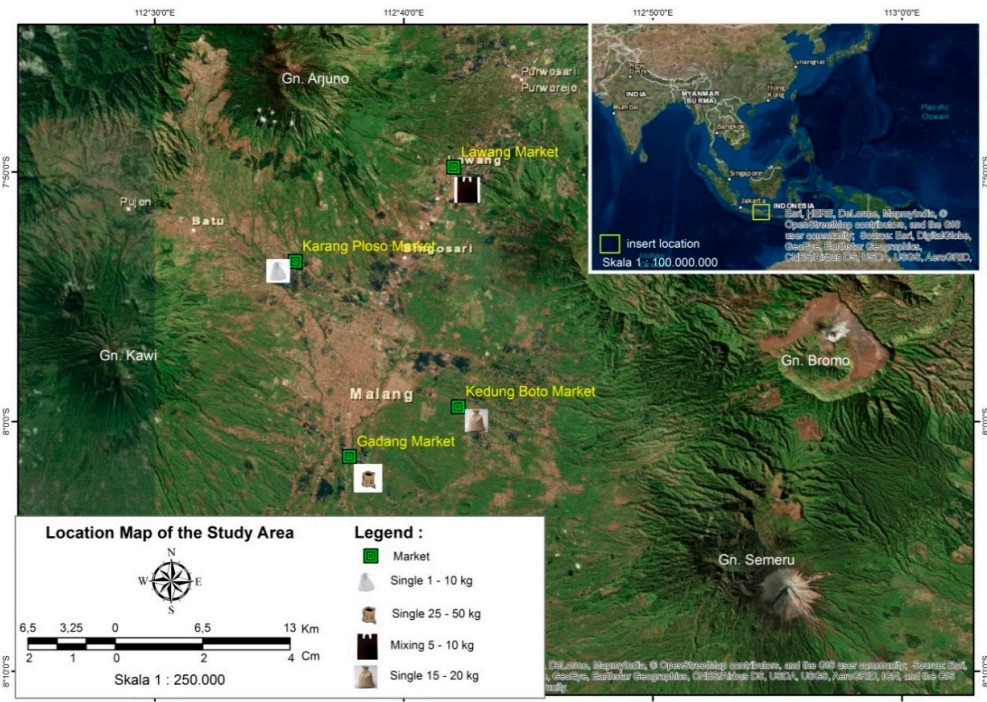

**Figure 1.** Research Location of Four Traditional Markets.

The observations were carried out in September 2016, January 2017 and July 2017. Each observation lasted for one week. The informants of this study consisted of 75 buyers, 20 distributors, as well as managers of the examined markets as key informants and 15 old traders, who witnessed the growth and development of the markets throughout the years. The primary data has been generated through a field-based ethnographic study from January 2014 to June 2015. Researchers conducted direct observations of the transaction activities in the market, as well as in-depth interviews with traders and buyers incidentally. Detailed interviews were conducted at the willing respondents' homes to obtain more complete data and impressions. Researchers also conducted visits on farms around the market within a 5 km radius to ascertain whether the working patterns of intermediary traders were correct as stated by the respondent in the previous interview. The researchers conducted this to ensure the first objective of research regarding how the market increases flexibility from time to time by changing various things starting from commodity initialization, changing market segmentation, operational time, reducing transaction costs, and changing transportation modes answered.

To complement the market's flexible data, the researchers also collected historical data by listening to the oral stories of village elders who lived around the market for more than 20 years. The data from the in-depth interviews were then appointed to focus group discussions (FGD) with elders around the market to describe the development of the markets over time and helped to clarify the data obtained during field interviews. This FGD is very useful to further analyze whether the field notes posted during in-depth interviews are true or not because FGDs can complement the data context which is usually relatively weak in interview sessions (Moretti et al. 2011). While exploring the market areas, five random elderly people were questioned and turned out to become the most important informants. Deep interviews were conducted to collect data about the merchants to keep transactions happening from the market opening to closing. This was conducted to ensure that all transactions and negotiation processes were well recorded.

In-depth interviews were conducted to explore the actors' strategy data to solve the problem of market coordination as the second main objective of this study. The in-depth interview is only guided by a list of simple questions about how the actors build a sale and purchase agreement, maintains the agreement, how to anticipate violations, and how the settlement mechanism was conducted when a violation occurred. Data collection was conducted by cross-checking respondent; verifying answers with field conditions, and visiting a location twice were the researchers' triangulation strategy. Data collection involved two researchers namely the first author and co-author who worked alternately during the interview. Cross checks were conducted after the interview process. These were conducted as part of the researcher triangulation, where the same problem is asked to the same person and determine whether there is a high variation or not (Kern 2018).

Data analysis using a qualitative content analysis method by conducting inter-theme mapping of field notes so that the narrative of the process of market change is institutionally constructed and the behavior of the actors is built (Cho and Lee 2014; Erlingsson and Brysiewicz 2017). Coding was done on interview texts to classify the themes, especially for the results of in-depth interviews then will be categorized and abstracted to draw the conclusions (Elo and Kyngäs 2008). In the process, of course, there will be a process of sorting data and simplification so that it becomes easier to do abstraction. While to strengthen qualitative content analysis theme analysis was used primarily to describe actor strategies dealing with coordination problems. Thus the researchers can use deductive and inductive strategies simultaneously (Fereday and Muir-Cochrane 2006) so that it is easier to quantify actor strategies.

## 4. Results

*4.1. The Flexibility of Market Institutions Amidst Competition*

### 4.1.1. Specializing Commodities

The first adaptation performed by the traditional vegetable markets is by specializing in certain commodities. At Kedungboto market, for example, the main commodities nowadays are spinach, kale and collards while in the 1990s, this market still sold a variety of vegetables and even kitchen utensils and groceries. From FGD with old traders, commodity specialization occurs naturally. At first, traders bought spinach, kale, and collards only as a complement to other wholesale goods. They stopped by at the Kedungboto market to complete the shopping in other main wholesale markets, such as the Blimbing market and the Gadang market. In the beginning, spinach, collards, and kale were second-class vegetables, after highland vegetables, such as cabbage and carrots, therefore the price was relatively cheaper. Over time, the demand for spinach, collards, and kale continued to increase and traders started to buy them from Kedungboto market in large numbers starting from the year 2000.

Some traders revealed that consumers liked collards, spinach and kale because the prices were relatively cheap compared with other types of vegetables. Besides, these vegetables are not seasonal, and the price is relatively stable throughout the year. The combination of availability and affordability has stabilized the demand and even tends to increase it. Almost all pitchman sell spinach, collards and kale as those are usually requested by the customers in higher amounts than other types of vegetables (Figure 2). Pak Tomo, a customer who had been shopping in Kedungboto market regularly over the past 20 years, revealed that *"More people go shopping in this market since it started selling only spinach, kale and mustard and the price became more affordable than at the other markets. Starting from the year of 2000*[sic]*, customers come not only from Malang but also from Surabaya and Bali."*

Commodity specialization also has a very close relationship with ecological farming areas around the market. Within a radius of 10 km around the market the area is very suitable for growing spinach, collards and kale and farmers can plant and harvest six to eight times during the year. Since the market is located closer to the farming areas, the vegetables are always fresh and the transport from the farms to the market causes little damage to them. The alluvial soil and the availability of adequate irrigation techniques, as well as the lifelong experience of the farmers have provided sufficient capital for the development of the commodity. Hence, specialization is not only driven by consumers' demand for certain vegetables, but also supported by agro-ecological advantages of the surroundings. State-promoted markets, on the other hand, sell types of vegetables that are not locally grown and therefore, are more expensive. The institutional strategy is built by a market creating efficiency, which is very closely related to the ecological conditions of the region and to government policies which "deliberately" weaken the presence of wet markets in Malang. The advantages of natural resources are easier for the market to specialize in the commodities they sell and build a kind of special cluster to increase the efficiency of their supply chain. (Ikram et al. 2018).

Another form relatively related to commodity specialization is branding the market as the center of a particular vegetable commodity, although other vegetables are also available. Sweet corn, for example, is known to be cheaper in Karangploso Market compared to Lawang, while Cucumber is the mainstay of the Nggadang market. Lawang market is famous as a place to buy the cheapest oyster mushrooms, as well as various traditional vegetables, such as bamboo shoots, ferns, and banana *ontong*. Market branding indirectly supports the institutional efforts to increase efficiency (Grashuis 2017) and for the long-term, it will increase market competitiveness itself. Branding is certainly very helpful for consumers in finding the vegetables they need, because each market has a certain commodity in stock and sold at a cheaper price. Therefore, the four markets in the research location identify themselves as a market for certain commodity specialists even though in practice their various agricultural commodities sell.

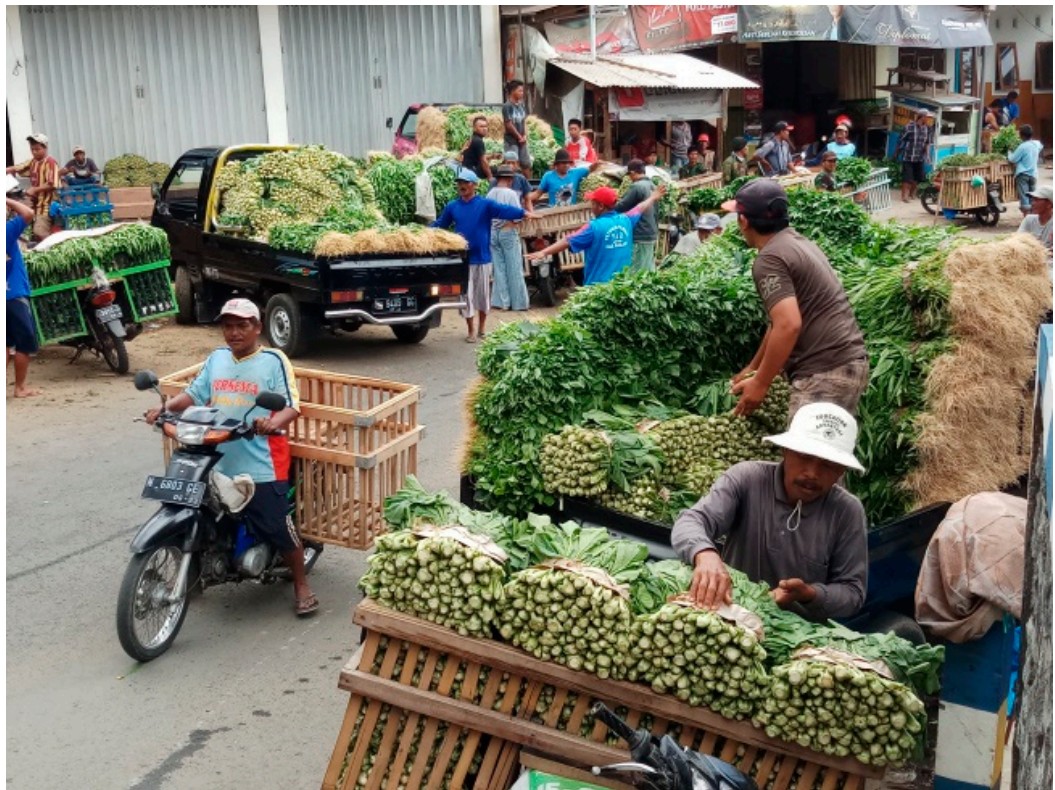

**Figure 2.** Spinach, Collards, and Kale at Kedung Boto Market.

### 4.1.2. Changing Modes of Transportation

The mode of transportation is adjusted by the traders to the harvest volume, limited road infrastructure, and fuel efficiency. With a harvest volume of only around two or three hundred kilos per farmer, transport by pick up car becomes inefficient. A motorcycle, on the other hand, can only transport up to one quintal (100 kg) and thus is inefficient, as well. Therefore, traders modify the motorbikes in such a way that they are able to carry up to three quintals of vegetables in one trip. Some traders also work as farmers. They sell their own rice yields or buy from neighboring farmers accordingly to the transport capacity towards the marketplace. The vegetables are tied together into bundles of fifteen to twenty kilograms, which maintains the quality of the vegetables and ensures a smooth transport across the narrow roads of the countryside. With such small-scale production, the most efficient means of transportation is a modified motorcycle.

Modified motorcycles are also an efficient mode of transport with regard to their capability to reach the lands that do not have adequate road access compare with pick up car. Rural infrastructure in third world countries can be relatively poor, which makes it difficult for heavy vehicles to pass through. With motors bikes, goods can be bought directly from the field (standing crop purchase) and be delivered freshly to the markets. Suwarno, a 45-year-old vegetable seller, said that *"people started using modified motorcycles in 1995 when the farmers planted their vegetables in small plots and the yield was decreasing gradually."* As an addition, the damaged and muddy road made it difficult for Suwarno to access the plantation/rice field. Even though he owns a pick-up truck, Suwarno keeps using the modified motorcycle because it is more efficient on fuel and he can arrive faster at the required locations. With regard to the small-scale productions, limited road infrastructure and fuel efficiency, modified motorcycles become the most effective means of transportation (Figure 3).

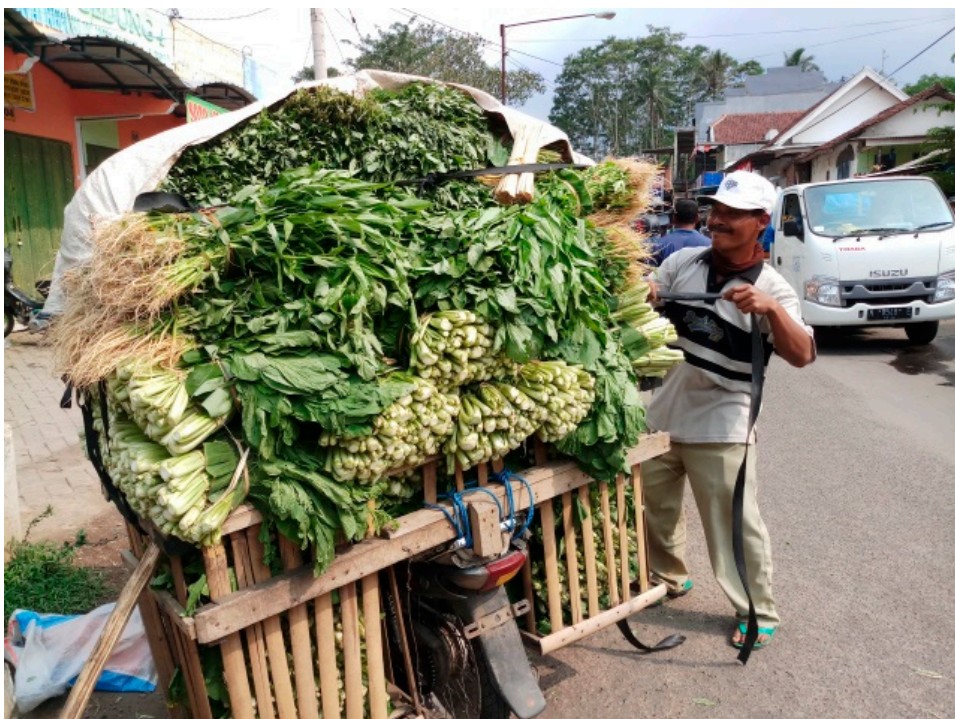

**Figure 3.** Modified Motorbike.

Because vegetable sources are rarely close to the market, the transportation tool also makes it easier for them to return and serve farmers offering fresh vegetables. A collector can go back and forth three times to carry goods. Cheap modification fees and low tax costs support these traders to change their transportation capital from the car to the modified motorbike. In addition, the splendid parking area in the traditional market area is natural in Indonesia, so the use of modified motorcycles is very important because it does not require too much parking space. The efficient and practicality are the main reasons for actors to use modified motors in response to poor and narrow road conditions, narrow harvest area, high mobility, to ensure accumulatively increasing institutional market efficiency. According to (Zant 2015) findings where transportation modes affect market prices so that transportation plays an important role in increasing efficiency.

### 4.1.3. Changing Market Operating Hours

Another adaptation mechanism traditional markets use to increase their competitiveness is to reevaluate their operating hours. Markets serving the wholesalers are open in the afternoon between 14 p.m. and 16 p.m. so that in the evening the buyers have time to sell to the markets serving the street vendors. Meanwhile, markets that serve restaurants and food stalls are open in the evening between 18 p.m. and 21 p.m. In addition, markets which serve to pitchmen are open from 1 a.m. to 5 a.m. in order to keep the vegetables fresh to be sold since they are taken around the residential areas. Characteristics of vegetables that are easily damaged and must be obtained by consumers in fresh conditions, the differences in the opening times of traditional markets make it easier for consumers to obtain vegetables according to their needs.

Kedung Boto market, for example, is open between 14:00 and 17:00, since it sells to the wholesaler who needs some time to re-pack the vegetables to meet the requirements of the end-users. The wholesalers usually pack the vegetables into a 10 to 20 km pack of vegetables and deliver them to retailers in smaller wet markets. Meanwhile, Karangploso market is open from 14:00 to 18:00, because it sells to small retailers who resell the vegetables in the morning to people living in residential areas. Lawang market opens at night and early in the morning because they sell directly to the retailers, as well as restaurant owners who prepare food for sale in the afternoon or in the evening.

These deliberate operating hours allow sellers to sell their produce in different markets. The short distances between the markets also make it easy for the sellers to go from one market to another in one day. Thus, the differences in operating hours of the markets give an opportunity for someone to shop in one market then resell the product in other markets. Since the operating hours for the state-promoted market are mostly limited to the morning and afternoon, the traditional market helps both sellers and customers to adjust their shopping time to their personal schedule.

Not only is it easy to divert unsold goods to other vegetable markets, but the time difference is also very helpful for traders to maximize vegetable packaging workers in the warehouse. Vegetable packaging workers can work all day from morning to night to increase productivity while streamlining salary payments. One trader can be a supplier in four different markets to ensure every vegetable can be sold, not merely the good quality ones. The difference in market operating hour not only facilitates the transfer of goods from one market to another, but also streamlines the work of packing workers. This efficiency will reduce the cost of producing vegetable packaging. This is in line with the findings (Purnomo 2018a), another paper in the same research project with this article, found that the flexibility of labor management greatly helped traders to enter several types of markets according to the quality of vegetables sold.

### 4.1.4. Creating Less Cost Management

One way to increase market efficiency is by reducing the potential transaction costs that arise in all transactions on the market (Jacques et al. 2018; Williamson 2010). The efficiency of transaction costs is carried out by minimizing expenses in the transaction process, such as loan interest, labor, and the cost of the scales. Transaction costs, such as parking fees, scales fee, and security costs are relatively inexpensive compared to the state-promoted market fees. The only fee is the regular payment to the village administration to be paid by the wholesalers. The small cost of the transactions has become one of the traditional market advantages compared with a state-promoted market, which has sets of charges for traders. Traditional market managers are very concerned about this problem because they realize that too many levies imposed on actors as extra expenses, the market will lose customers. On state promoted markets, formal and informal levies, such as tax, parking, and renting a place, often occur because local governments need tax revenues (Figure 4).

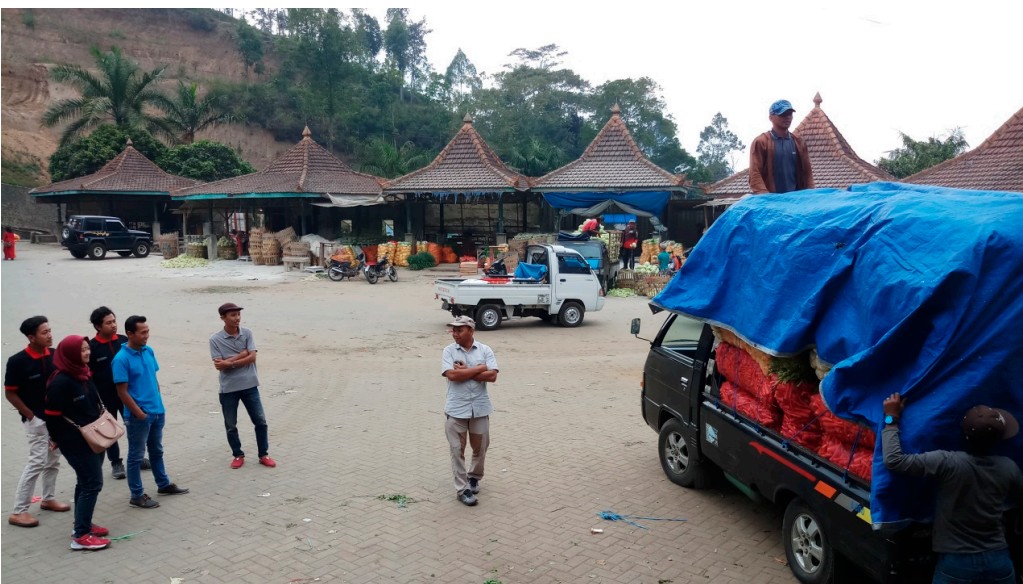

**Figure 4.** State promoted market activities.

The market also provides opportunities for actors to get cheap working capital—not only by avoiding levies, but through wholesalers who provide loans to traders with low to no interest. The debt

is one of the instruments used to maintain the loyalty of the collector. Traders take risks by providing loans without collateral and without interest. Mr. Samino, a 63-year old seller who has been selling produce for more than 15 years revealed that *"loan is a common practice in this market. No collateral is needed because you trust people whom you lend money to. Everyone has to keep each other's trust so that this practice shall continue. There are sometimes one or two people who do not pay their loan."* The ease of obtaining working capital is inseparable from the punishment mechanism, which is hard for borrowers who do not return loans. These violations are responded with difficulties in making transactions in the market due to the existing stigma.

The small costs issued by market participants could not be separated from the business scale of traders. By selling only two to three quintals of goods, they do not release the cost of labor. Collectors harvest their own crops in the fields with the help of farmers. Even traders can pay back to farmers so that they are not burdened with capital adequacy. Farmers give two to three days after harvesting without collateral. Wholesalers themselves directly pay cash for the vegetables they sell. On the other hand, state promoted market is still burdened with additional labor costs and rental space, which burdens the small merchants. This also makes traditional markets the choice of traders because official tax and illegal levies are minimized.

Other costs, such as porters, do not apply as well, while it exists in the market made by the government. On the market initiated by the government, traders should need to hire carriers and scales even though they are only selling one quintal of vegetables. Traders raise their own goods and wholesalers are allowed to use their own scales in the transaction. In state promoted markets, sellers have to hire a scale that is usually monopolized by the manager of the market, thereby increasing transaction costs. Therefore, synergistically, markets create efficient transaction mechanisms so that the profits are not eroded by the costs of additional transactions (Carey et al. 2011). It is different from the state promoted market in which prices become the major incentive; the traditional market emphasizes developing a social network. The situation above shows those traditional markets are very flexible avoiding additional costs that often appear on the market institutionally in facing competition with state promoted markets.

## 4.2. Market Actor Strategies on Overcoming Coordination Problems

### 4.2.1. Overcoming Competition Problems

If the above discussion deals with the markets' performance, the problem of coordination focuses more on social interaction problems of actors within the market. One determinant of the sustainability of the market in maintaining its existence is the active role of the actors to resolve issues of coordination, namely the problems of competition, valuation, and cooperation issues (Beckert 2012). To overcome the problem of competition, traders actively build social relationships with the suppliers by utilizing local traditions, such as attending weddings, circumcisions, or other traditional feasts. If they cannot make an appearance at the party, they will leave it to other traders so that the relationship is maintained. By building social relationships, the traders can reduce the level of competition because transactions do not solely rely on economic considerations, but also happen for sociological and emotional reasons (Çalişkan and Callon 2010).

Besides, the wholesalers often visit the suppliers if they, or their families, are sick. They also actively exchange greetings or ask for news about fellow traders. Text messages and social media become important communication tools to strengthen the relationships between wholesalers with the suppliers. Some wholesalers even give presents to the suppliers every year on the day of Eid Al-Fitr, such as food parcels, shirts, and cash. The closeness of their social relationships maintains their loyalty. Although the prices in other markets increase, the traders still send goods because of their friendships. They are often compromised by selling goods to other markets with better prices, but still sell goods to traditional markets to maintain social relationships. Despite a declining volume, the market will still get a steady supply of goods even when the supply or production of vegetables from farmers is low.

If there are traders who do not sell goods to wholesalers for a long time, the issue will be discussed among fellow traders. This pattern is effective to keep the loyalty of traders, as well as to recognize the character of their customers. A bond of debts is also used as an instrument for wholesalers to tie customers' loyalty. If they had debts, it would appear a shame if they would not sell goods on a regular basis. With such social ties developed among actors, the market will still get a supply of vegetables, even though the purchase price is low. The flexibility thereby improves the durability of the market. Even with quite extreme price disparities, traditional vegetable markets still get supplied with goods. If the condition of the market starts to become unstable and fewer goods enter the market, a new price will normally be created. Social capital, and its synergy with the adaptation strategies of other markets, is proven to be an important factor for the sustainability of the market.

Market actors will try to reduce competition by building relationships in emotional level through various processes of social transactions. The use of personal sentiments is very useful to reduce competition where actors will perform "get action" and "blocking action" on the limits of fairness to obtain profits. This means that competition is a space for actors to ensure the value of goods in the market is dynamic. Therefore, there is a difference in the cost of production as a profit to be taken. Without this competition, there will be an equilibrium process. There would be no price difference. Hence it will provide revenue to the actors. On the other hand, large revenue would lead to no price agreement. Therefore, solving this competition problem will be the trigger for the process of forming the value of goods traded on the market (Beckert 2009).

### 4.2.2. Overcoming Valuation Problems

The sustainability of the market is also determined by the ability of the actors in agreeing on the prices of goods so that a transaction can occur. The price must be competitive of course, so the transaction remains mutually beneficial and satisfies both, buyers and sellers. The price is basically an instrument to accommodate the interests of market participants (Fligstein 1996; Dekker 2016). In Kedungboto market, as an example, sellers prefer to take small profits with high volume instead of big profits with yet a small volume. This principle is proven very helpful to ease the price formation.

This principle is also very helpful to maintain price stability. Compared with the market established by the government, traditional markets are relatively more stable, hence the traders feel more secure when making transactions. According to traders, relatively stable prices allow them to set higher prices. So, traditional markets are able to facilitate the formation of the value of goods more easily and approach a more objective price according to the quality of goods because the sellers are willing to reduce their ego to get higher profits. Adaptation, trust and commitment are identified as key drivers for value creation (Walter and Ritter 2003).

The ease of price formation has become the main condition of the transaction. The actors always compare prices in traditional markets with other markets that are more expensive. It resolves the valuation problem where price formation occurs based on an objective assessment process by comparing prices of the same goods. This classification may be based on standards that make it possible to objectively describe the quality of products in relation to other products of the same class (Beckert 2007; Callon 1998). This situation reflects Callon (1998) idea that actors in the market are calculative agencies who have value, culture, rules and passions instead of sole "atomic" economic goals. These individuals are naturally formatted, framed and equipped with prostheses, which help them in their calculations, as well as agents who actively contribute to developing market configurations.

Not only the price, but also the availability and quality of goods, cause transactions in the traditional market in term of valuation process to be difficult. The traditional vegetable market lets the buyer classify in a flexible way and consider the commensuration of the prices, in which actors rank products according to their contribution in the fulfillment of a functional need within a status or order of goods. According to the traders, they have many quality options to make it easier choosing the goods based on the tastes of their customers. With the price comparison between the sellers,

the buyers will choose accordingly to the characteristics of their customers. If they sell the vegetables to a province's market (Surabaya or Porong), they can buy collards, kale, and spinach in all kinds of shapes. However, if they sell it to retail markets in Malang, the vegetables should be uniform, since they will be divided into small packs and sold like that.

Almost in the four markets researched are very dynamic in determining prices. Each actor only takes a small difference from the cost of production to ensure relatively easier price determination. This occurs as the previous market has institutionally carried out the process of efficiency both transportation and reducing transaction costs to ensure lower production cost. The low production cost makes the difference between the cost of production and price as the potential profit contested in the transaction process. Smaller difference due to institutionally inefficient market would complicate matters in establishing prices.

### 4.2.3. Overcoming Cooperation Problems

Cooperation problems can occur, if one of the actors commits fraud—such as an incompatible price-quality condition that formerly had been promised. If this problem is not resolved, it may cause the valuation problem had not resolved that potentially upset market instability. Transactions occur only if the actors agree upon the value of the goods based on their quality. Buyers will engage in market exchange if they are confident that the seller will deliver a certain quality of goods made in accordance between cooperation partners in the market. In traditional markets, the buyers already have "reputation of partners" or a list of traders who cheat and those who do not show a certain moral standard for an individual (Fourcade and Healy 2007; Pirson et al. 2017).

Sellers will provide actual information about the quality of goods, such as its freshness, age, and level of damage, to make buyers feel comfortable. This phenomenon is called "reputation effect" or the effort of actors to preserve their reputation in order to maintain social relationships (Podolny 2010). The sellers very rarely manipulate the quality of their products by mixing grade A and grade B vegetables, or by tucking bad vegetables in the middle of a vegetable bond. Usually, the buyers check only briefly what they are actually buying. Therefore, a mutual trust among them is a prerequisite for the cooperation to continue (Podolny 2010). In this regard, the buyers are confident that the goods are bought based on this social contract, where trust is the fundamental precondition of stable market relations (Jraisat 2016; Pedroza-Gutiérrez and Hernández 2017).

If price and quality are non-conformant, or there is the seller that violates the agreement, the buyers usually speak satirically to the traders. Satire is usually responded by traders with apologies and the provision of technical reasons for the low quality, such as bad harvesting or damaged goods during transportation. To anticipate the possibility of fraud, there are social mechanisms, such as the effective sanctioning of defectors. Sanctions are not done directly by disconnecting trading relations, but gradually based on the degree of guilt. If the buyers have a major complaint, then the seller will provide compensation in form of lower prices or hand out certain bonuses for future purchases. Sellers are interested in the maintenance of trade relations and see to it that those ties are not broken because of the violation.

The mechanism of individual sanctions against violation of agreements is very effective in reducing problem cooperation. Each party will try to improve its social ranking to be more "trustworthy" in order to join in market interactions. This effort will certainly be carried out by all market actors without exception because if they continue to engage in fraudulent activities it can result in throwing off transactions until they could no longer trade. Recovery from trauma due to cheating takes a long time, it will even continue to be recorded by market participants and socialized continuously to each other. Bankruptcy reputation recovery process runs easier than error due to moral hazard. Traditional markets use this method to select actors who overcome cooperation problems, to ensure nothing disturbs market stability.

## 5. Discussion

Flexibility in the given context refers to the ability of traditional markets to naturally develop strategies in order to overcome competition among traditional markets and increase their internal efficiency. Gadang market, for example, which used to deliver to major vegetable vendors and retail buyers in the Malang region, ceased to exist in 2017. Lawang market specifically supplies food stalls and restaurants, as well as street vendors. Meanwhile, Buring market only supplies pitchman. Karangploso market only delivers to traders who will resell at traditional markets or vegetable stalls in residential areas. Customer segmentation proved to be successful in maintaining the markets' existence and reducing competition among them. At the same time, by specializing in the sale of certain commodities new customers can be attracted, thus generating an increasing number of transactions in a market economy. Such conditions indicate that both, the actors and the traditional markets as social institutions, possess high flexibility rates in order to adjust to changes in consumer behavior. The flexibility of traditional markets above is proven to increase market efficiency to ensure contested profit margins become larger than other markets. It would then strengthen market competitiveness in meeting changing consumer demands.

The growth of technology and improvement in rural transportation system encourage the flexibility of actors, as well as markets, allowing the traditional markets to survive business competition. It shows that, besides infrastructure, market devices, such as mobile phone and motorbike, are another significant factor to determine social interaction in the markets (Muniesa et al. 2007). Spreading information about prices from other markets allows actors to identify competition and make a quick adjustment. Therefore, changing in means of transportation reduce the risk of damaging goods, reduce conveyance cost, and the goods reach their destination faster. Thus, a naturally developing flexibility is useful for markets as an institution, and supported by market devices. This exhibits that the traditional market in research locations as social institutions has high flexibility by utilizing the opportunities and available technology. With clear segmentation, and openness in the use of technology, it is able to increase market efficiency institutionally to increase competitiveness. This situation is consistent with (Jackson 2015) stating that a strong market is determined by its capability to change values, consumer segmentation, commodity changes, and social interaction changes, among others.

Simultaneously, institutional changes in the market are also accompanied by the behavior of the actors who try to maintain the transaction process by ensuring that the three main problems of the coordination are resolved. In a more detailed elaboration, competition problems are resolved by utilizing an emotional sentiment of friendship social relations; the formation of prices through effective networks of information; and the cooperation problem through dealing with building a system of punishment and reward based on informal mechanisms. This proves that actors in traditional markets have the ability to solve coordination problems by utilizing social capital in their economic interactions. Through a high social capital, the customers' loyalty is increased, and dishonest behavior is prevented, thereby making the market transaction cost more efficient. This is in line with Kananukul et al. (2015), who state that trust as a component of social capital plays an important role in facilitating efficiency amidst various information uncertainties so that transactions can continue (Kananukul et al. 2015).

In comparison to e.g., farmers' markets in the USA, traditional markets in developing countries rely solely on internal efficiency, building social capital and making use of their ecological advantages. The traditional markets in Western countries depend on consumer perception towards goods they sell, and ethics attached to certain commodities, such as the support of local farmers. The given examples in this paper provide new insight for customers that several markets did very little to adjust to changes in customer behavior certainly state promoted in the market. On the other hand, traditional markets are very active and flexible in dealing with the competitive environment. Due to government campaigns that encourage people to shop in modern markets instead of traditional ones, studies from consumers' points of view will provide some insights into the changing face of traditional markets. Traditional market is capable to change in institutional manner and actors' behavior. It indicated that traditional markets still have strong competitiveness, not only their ability to build efficient market institutions,

but also the ability of their actors to solve coordination problems very effectively. This condition verified the researchers' initial opinion that the traditional market had good competitiveness by fully utilizing the advantages of social capital between actors that they had built so far in competing with the state promoted markets.

According to the Porter five forces analysis approach, the findings above shows that new market players are relatively difficult to enter traditional markets because they must have social relationships with buyers. A buyer will select both the integrity and capability of new entrants, to ensure the stability of their supply. Whereas, the suppliers have a strong bargaining power because they can divert sales to other markets, or terminate sales. This can be a reason for them to increase selling prices, especially if demand rises or supply decreases. While, due to the small number of buyers, the dependence of the sellers is relatively large that they have the power to demand low prices and good quality. Supported by the existence of social sanctions if the sellers commit fraud, it gives space for buyers to pressure their suppliers. Meanwhile, for sellers or buyers because the process of market specialization is very difficult to substitute goods so that this strategy cannot be used by both of them to move commodities. The fifth force, which is rivalry among existing competitors, in the case of traditional vegetable markets is relatively small because the relationship that is built is not only based on economic interests, but also emotional relationships so that this rivalry is relatively easy to avoid because it will potentially break established social ties.

## 6. Conclusions

Facing competition with modern markets formed by the government, traditional markets adapt well on an institutional level, as well as individually, in regards to the actors. Institutionally, commodity specialization is pursued to gain ecological benefits and to build a brand. Adaptation is also performed by using more appropriate forms of transportation in which considering the dynamic and precision of the vehicle becomes a more important aspect than speed and the ability to transport larger volumes. Another way of adaptation is the reduction of transaction costs by providing free parking, security, and avoiding other expenses. Additionally, giving and taking loans is rarely burdened with interest charges. Giving a loan is interpreted by traders as a form of investment to maintain customer loyalty. Furthermore, costs for porters, scales, and warehouse fees are not included in the transactions. Simultaneously, the whole process of adaptation leaves an impact on the high efficiency of Indonesia's traditional markets. As stated by Jackson (2015), the market as an institution entails a certain flexibility in order to make an adjustment when facing both internal and external problems.

This institutional adaptation not only targets the transaction cost efficiency, but also the efficiency of production costs. This is very beneficial for traditional markets because the average promoted market state relies on efficiency by increasing the volume of goods, where the greater the volume of goods sold, the more efficient it would be. With a narrow farmers' land ownership, getting a large number of vegetables in a short time is difficult, because the area of vegetable cultivation is narrow on average and easily damaged. Traditional market flexibility in creating efficient economic transactions while minimizing the cost of efficient production proved very effective in increasing their competitiveness.

Meanwhile, when dealing with coordination problems to uphold the sustainability of the market, actors in traditional markets fully exploit their social capital thought with a close relationship between sellers and buyers helps the formation of prices, reduces competition, and facilitates the coordination process. A high degree of trust towards the compliance with previous agreements encourages the actors to maintain these relationships. By doing so, the problem of competition among market actors can be overcome and value creation and cooperation as a prerequisite of market sustainability are based upon a mutual understanding. In summary, traditional wet markets in Indonesia apply two strategies to guarantee their sustainability both become more flexible and efficient on an institutional level and maintain effective coordination activities between the actors (Shepherd 2004).

Market flexibility as a social institution adaptation to external development has proven to be very effective in increasing efficiency and competitiveness. Market actors dynamically build strong social capital to solve coordination problems as a condition for transactions on the market. The combination of institutional changes and actor dynamics in solving coordination problems simultaneously can increase market competitiveness to ensure survival despite the pressure of state promoted markets. If associated with Porter's five forces analysis, the coordination problem will be threatened by two components, which are bargaining of power from sellers and buyers, while other aspects, such as competition, substitution, or threat to entry have no potential to affect. Hence, the market will be maintained if each actor wants to reduce his bargaining power then the transaction takes place. Based on the above explanation, further research needs to focus on how much efficiency is built based on market flexibility, in terms of the efficiency in the transaction cost, and the efficiency of production at a more quantitative level. Future research may focus on the forms of actor strategy, based on rational or pure economic choices or moral or non-economic considerations, in solving coordination problems.

**Author Contributions:** This research does not involve enumerators or other researchers. Therefore, the findings are the research result conducted by the M.P. and F.O. The H.F. designed a research and data analysis theme and strategies, as well as formulated recommendations to ensure field research.

**Funding:** This research received no external funding.

**Conflicts of Interest:** The authors declare no conflict of interest.

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
