# Peer review of "Indonesian Traditional Market Flexibility Amidst State Promoted Market Competition"

_socsci, doi:10.3390/socsci7110238_

Round 1

Reviewer 1 Report

In the assessment of the paper submitted for the review, I specifically focused on the discussed issues, applied research procedure, substantive content of the paper and its structure.

The considerations conducted in the paper are focused on such categories as:

Indonesian traditional market, state promoted market, coordination problems, market flexibility.

The subject area discussed in the paper should be considered important and topical. It is also consistent with the profile of the Journal.

However, deliberations conducted in the paper need to be expanded.

The structure of the paper is not clear (recommended structure: abstract, introduction, review of literature, methodology, results, discussion, research limitations, managerial implications, conclusions).

It is specifically recommended to:

- clarify the purpose of the paper (n the abstract of paper),

- develop description concerning materials and methods,

- specify the managerial implications(as a separate part of article), 

- indicate the research limitations and formulate trends for future research.

Author Response

Dear Reviewers,

I send several clarification to improve our manuscript. We give more detail explanation in background, methods, result, discussion, and conclusion including strategies for future research. I hope it would be met with the journal standard. Thank your for your attention.

Best Regard,

Reviewer 2 Report

Reviewer Comments on Socsci-381935

This paper illustrates the continued functioning and survival strategies of traditional markets and street vendors in Indonesia in the face of competition from state-promoted markets. The writing style is vivid.  References to the literature, however, are given to support the key points or to make comparisons.

 I have a few questions and comments that need to be addressed.

1) I would suggest expanding Section 4 on methods. Please provide further details on this ethnographic study. What is the main objective of the study? How many researchers are involved? How is it financed? What other (if any) research output is produced to date (articles, reports, working papers, etc). Please also provide more information on the interviews, questions asked, typical length of an interview, descriptive statistics on the interviewees, etc. Overall, you need to provide further documentation on the methodological side.

2) Since the study is on the competitive pressures on traditional markets, an outside reader would want to know more on the other side of the competition. For instance, how did the market shares evolve overtime? Could you take some goods as examples and provide numbers?

3) Linked to the question above, there might be some survivorship bias in the overall results. Just looking at the practices of "successful" companies, entrepreneurs, or street vendors consists of such a bias since it ignores those less successful ones (or, those that simply went out of business or left the market to do something else). Can you please elaborate on how you deal with this issue in your study?

4) I would suggest you to include a SWOT and Porter's Five Forces analysis for the traditional markets and street vendors in your study. You have many of the ingredients for these analyses in your paper already and it would be good to present them in a more formal framework.

Author Response

Dear Reviewers,

Thank you for your comment on our manuscript. We already improve the manuscript base on your comment. But, we not really agree with your suggestion to employ SWOT and  Porter's Five Forces analysis in our analysis. Those theories base on market management tradition, whereas our manuscript base on market sociology tradition. For more detail explanation, we enclosed the clarification along with this letter. Thank you for your attention.

Round 2

Reviewer 1 Report

Thank You for introducing my suggestions in article.

Author Response

Dear Reviewer of Social Science Journal,

I am very happy that you already accept our revision manuscript. according to your suggestion, I will ask English to MDPI English assistance to improve our manuscripts.  Thank you for your attention. 

with best Regard,

Reviewer 2 Report

Dear Authors:

Thank you for your efforts to revise your paper in line with my and other reviewers' suggestions. I am satisfied with the changes made and the explanations provided. I will not insist on SWOT and the Porter's Five Forces framework. (I myself studied both economics and sociology and I see your point. But, a cross-disciplinary approach could still be beneficial.)

You included some pictures from the interviews in your reply to me. It would be good for third readers to include them in an Appendix. It would complement the vivid narrative in the main text well.

Best Regards...

Author Response

Dear Reviewer Social Science Journal,

We finished our revision base on your suggestion. figure of our research activities include in the manuscript as well as your suggestion on porter's five forces analysis. We hope it pass the article standard. Thank your for your review and attention.

With kind regard,